# Predictive Value of Urinary Aquaporin 2 for Acute Kidney Injury in Patients with Acute Decompensated Heart Failure

**DOI:** 10.3390/biomedicines10030613

**Published:** 2022-03-06

**Authors:** Ming-Jen Chan, Yung-Chang Chen, Pei-Chun Fan, Cheng-Chia Lee, George Kou, Chih-Hsiang Chang

**Affiliations:** 1Kidney Research Center, Department of Nephrology, Chang Gung Memorial Hospital, Linkou Chang Gung Memorial Hospital, Taoyuan 333, Taiwan; b9202066@cgmh.org.tw (M.-J.C.); cyc2356@cloud.cgmh.org.tw (Y.-C.C.); 8802019@cgmh.org.tw (P.-C.F.); a12490@adm.cgmh.org.tw (C.-C.L.); mr8393@cgmh.org.tw (G.K.); 2Graduate Institute of Clinical Medical Science, College of Medicine, Chang Gung University, Taoyuan 333, Taiwan

**Keywords:** acute kidney injury, acute decompensated heart failure, coronary care unit, urinary aquaporin 2

## Abstract

Acute kidney injury (AKI) is frequently encountered in people with acute decompensated heart failure (ADHF) and is associated with increased morbidity and mortality. Early detection of a urinary biomarker of kidney injury might allow a prompt diagnosis and improve outcomes. Levels of urinary aquaporin 2 (UAQP2), which is also associated with several renal diseases, are increased with ADHF. We aimed to determine whether UAQP2 predicted AKI in patients with ADHF. We conducted a prospective observation study in the coronary care unit (CCU) in a tertiary care university hospital in Taiwan. Individuals with ADHF admitted to the CCU between November 2009 and November 2014 were enrolled, and serum and urinary samples were collected. AKI was diagnosed in 69 (36.5%) of 189 adult patients (mean age: 68 years). Area under the receiver operating characteristic curve (AUROC) of biomarkers was evaluated to evaluate the diagnostic power for AKI. Both brain natriuretic peptide and UAQP2 demonstrated acceptable AUROCs (0.759 and 0.795, respectively). A combination of the markers had an AUROC of 0.802. UAQP2 is a potential biomarker of AKI in CCU patients with ADHF. Additional research on this novel biomarker is required.

## 1. Introduction

Acute kidney injury (AKI) is common in individuals with acute decompensated heart failure (ADHF), and several subtypes of cardiorenal syndrome were proposed in 2008 [1]. More than 20% of patients with ADHF develop AKI during hospitalization, and AKI in such patients is associated with an increased risk of mortality [2,3,4]. The severity of AKI is also associated with the degree of ADHF. However, the diagnosis of AKI by using the creatinine method is imperfect due to its 24–72 h delay in elevation from onset [5]. Clinicians may deploy nephroprotective measures to improve the outcome of patients with ADHF if AKI is recognized early. [6] Several biomarkers have been proposed to detect AKI in this scenario, such as the brain natriuretic peptide (BNP), cystatin C, neutrophil gelatinase-associated lipocalin, kidney injury molecule-1, proenkephalin, and urinary TIMP-2 × IGFBP7 [7,8,9,10,11,12,13]. Although some biomarkers have promising results and have been validated in several clinical trials, implementation trials are slow to emerge, possibly due to heterogenous results among biomarkers studies [14,15,16]. The Kidney Disease: Improving Global Outcomes conference urged new studies evaluating the diagnostic, treatment-guiding, or prognostic role of biomarkers [16].

Aquaporin 2 (AQP2) is a water channel present mainly in the principal cells of collecting ducts. It is regulated by arginine vasopressin (AVP) and responsible for regulating urine concentration [17,18,19]. AVP binds to the AVP type 2 (V2) receptor, causing apical trafficking of intracellular AQP2 through cyclic-adenosine-monophosphate-dependent phosphorylation [20,21]. Conversely, the V2 receptor antagonist tolvaptan prevents AQP2 trafficking and decreases urine osmolality [22]. Approximately 3% of AQP2 in the kidney is excreted in urine; AQP2 in urine can be deemed a noninvasive marker of collecting duct responsiveness to AVP [23,24]. Urine AQP2 levels are increased in several clinical conditions, such as heart failure, the syndrome of inappropriate secretion of antidiuretic hormone, cirrhosis, and pregnancy [25,26,27,28]. In addition, UAQP2 levels are increased in diabetes nephropathy and are a potential noninvasive biomarker in predicting the clinical stage [29]. Furthermore, changes in urine AQP2 levels have been observed in several animal models, including ischemia–reperfusion (I/R), cisplatin-induced, and gentamicin-induced AKI animal models [30,31,32]. Given the potential diagnostic role of AKI, this study was aimed at determining whether urine AQP2 can predict AKI in patients with ADHF.

## 2. Materials and Methods

We conducted a prospective, observational study in the coronary care unit (CCU) of a 3700-bed tertiary care referral center in Taiwan between November 2009 and November 2014. Patients diagnosed as having ADHF were enrolled. We included a total of 189 patients and divided enrollees into AKI and non-AKI groups. We excluded patients who had a baseline estimated glomerular filtration rate (eGFR) of <30 mL/min/1.73 m^2^, were receiving kidney replacement therapy, were aged <18 years, or had reported any prior organ transplantation. No enrolled patients had been exposed to vasopressin V2 receptor antagonist such as tolvaptan during admission. The study protocol was approved by the local institutional review board (no. 201401993B0). We prospectively collected the following data: demographic characteristics, routine hemogram and biochemistry test results, and hospital outcomes. Biochemistry and hemogram values were measured by the central laboratory of Chang Gung Memorial Hospital.

The diagnosis of ADHF was based on European Society of Cardiology criteria [33]. All the patients received standard medical therapy for ADHF. Both AKI and non-AKI group patients received standard medical therapy for ADHF based on CCU clinician decision. Our aim was to determine the predictive value of UAQP2 for AKI. The primary outcome was any stage of AKI within 7 days after admission to the CCU. AKI was defined as either an increase in SCr by ≥0.3 mg/dL within 48 h or an increase in SCr to ≥1.5 times the baseline within 7 days, according to the definition in Kidney Disease: Improving Global Outcomes (KDIGO) Clinical Practice Guidelines for Acute Kidney Injury. The severity of AKI was also judged according to the KDIGO guidelines. Secondary outcomes were 180-day and 365-day mortality. We also followed up on participants for 12 months by reviewing electronic medical records or conducting telephone interviews.

Urine samples were collected in sterile nonheparinized tubes immediately after admission into the CCU. Collected samples were centrifuged at 5000× *g* for 30 min at 4 °C to remove cells and debris. The clarified supernatants were extracted and then stored at −80 °C until further analysis. UAQP2 was measured using the ELISA Kit for Aquaporin 2 (Cloud-clone Corp product number SEA580Hu, Katy, TX, USA). The test protocol adhered to the manufacturer’s specifications.

Continuous variables (i.e., age and laboratory data) of the AKI and non-AKI groups were compared using the independent samples *t* test. The biomarkers of interest (serum BNP, UAQP2, and UAQP2/urine creatinine (UCr) were compared using the Mann–Whitney *U* test due to a lack of normality. Categorical variables, including outcomes, were compared using Fisher’s exact test. The trends of UAQP2 and serum BNP across AKI stages were assessed using the Jonckheere–Terpstra trend test. The association between the biomarkers of interest and the risk of AKI was investigated through logistic regression analysis. Several well-established risk factors of AKI and HF were adjusted in the multivariable logistic regression model, including age, sex, diabetes, hypertension, mean atrial pressure, left ventricular ejection fraction (LVEF), hemoglobin, and baseline serum creatinine.

Area under the receiver operating characteristic curve (AUROC) was used to examine the discrimination abilities of the biomarkers in diagnosing AKI. We further compared the AUROC of BNP alone to that of BNP plus UAQP2 and that of BNP plus UAQP2/UCr. The standard error of the AUROC was calculated using DeLong’s nonparametric method. The composite outcome of AKI stage 3 and in-hospital mortality was also analyzed, in addition to AKI stages 1–3. Last, according to the optimal cutoffs of UAQP2 and UAQP2/UCr determined by the Youden index, we compared the 180-day survival rates of the higher and lower subgroups by using the log-rank test. A two-tailed *p* value of less than 0.05 was considered statistically significant in our study. We used SPSS 25 (IBM SPSS Inc, Chicago, IL, USA) for data analyses.

## 3. Results

### 3.1. Patient Characteristics

Overall, 189 adult patients (129 male and 60 female) were investigated. AKI was diagnosed in 69 (36.5%) patients. Compared with patients in the non-AKI group, those in the AKI group had the following characteristics: they were older, were less likely to have LVEF and more likely to have chronic kidney disease (CKD) or hypertension, and had higher baseline serum creatinine, lower hemoglobin, and higher potassium levels (*p* < 0.05). Regarding the biomarkers of interest, the median serum BNP levels were 1210 pg/mL and 479 pg/mL, and the median UAQP2 levels were 61.5 ng/mL and 30.9 ng/mL, respectively, in the AKI and non-AKI groups (*p* < 0.001). To compensate for variation in urine dilution, urine UAQP2 values were adjusted according to UCr. The median UAQP2/Cr levels in the AKI and non-AKI groups were 1.09 fmol/mg and 0.35 fmol/mg, respectively (*p* < 0.001; Table 1). In addition, levels of both serum BNP and UAQP2 increased with AKI severity (Figure 1).

Approximately half the AKI patients had stage 2 or 3 AKI (Table 2). Eventually, seven (11.1%) of the AKI patients underwent hemodialysis. A total of 24 patients had the composite outcome of AKI and in-hospital mortality. Patients in the AKI group had a significantly longer hospital stay. The patients also tended to have higher in-hospital mortality, higher readmission within 180 days, and higher readmission within 365 days than their non-AKI counterparts, although the difference was nonsignificant (Table 2).

### 3.2. Association of Serum BNP, UAQP2, and the Risk of AKI

When known risks of AKI were not adjusted for, higher serum BNP, UAQP2, and normalized UAQP2 were significantly associated with an increased risk of AKI (Model 1 of Table 3). These biomarkers remained significantly associated with the risk of AKI despite further adjustments for all covariates (Model 5 of Table 3). By contrast, the association between these three biomarkers and the risk of the composite outcome (AKI stage 3 and in-hospital mortality) was not significant with adjustment for covariates.

### 3.3. Discrimination Abilities of BNP and UAQP2 in Detecting AKI

The AUROCs of serum BNP, UQAP2, and normalized UAQP2 were 75.9% (95% confidence interval (CI) 69.0–82.9), 79.5% (95% CI 73.2–85.7), and 76.1% (95% CI 69.1–83.2), respectively, with acceptable performance (Figure 2a). The AUROCs of BNP plus UQAP2 and BNP plus normalized UAQP2 were 80.2% (95% CI 74.1–86.4) and 80.2% (95% CI 73.9–86.5), respectively, which were slightly greater than that of BNP alone (Table 4). In addition, the analysis of composite outcome is also provided (Figure 2b and Table 4).

### 3.4. 180-Day Survival Rates of High- and Low-Biomarker Subgroups

The optimal cutoffs of UAQP2 and normalized UAQP2 were >35.3 ng/mL and >0.83 fmol/mg, respectively (Table 4). However, whether UAQP2 was normalized or not, no significant difference in the 180-day survival rates of the high versus low subgroups was observed (Figure 3).

## 4. Discussion

To our knowledge, this is the first study to evaluate UAQP2 levels in AKI among CCU patients with ADHF. Regardless of whether they were normalized by UCr, UAQP2 levels were significantly higher in those with AKI than in those without. In addition, UAQP2 levels increased with AKI stage. Even after age, sex, diabetes mellitus, hypertension, mean arterial pressure, LVEF, and baseline creatinine were adjusted for, the UAQP2 level was associated with the risk of AKI. UAQP2 also demonstrated fair discriminating power for AKI and the composite outcome of stage 3 AKI and in-hospital mortality. Our findings support the diagnostic potential of UAQP2 in AKI for individuals with ADHF.

Neurohormonal activation plays a key role in cardiorenal syndrome. As heart failure becomes more severe, blood pressure lowers and renal perfusion worsens, causing baroreceptor and renin–angiotensin–aldosterone system activation, respectively [34]. Neurohormonal activation further worsens already impaired cardiac function, which in turn leads to further deterioration of target organ function, forming a vicious cycle [35]. Among the complex neurohormonal activation pathways, AVP plays an important role in heart failure [36]. In patients with heart failure, decreases in effective circulatory blood volume paradoxically cause AVP increase [37,38]. In the kidney, AVP binds to the V2 receptor, leading to increased production of cyclic adenosine monophosphate, further causing AQP2 phosphorylation via protein kinase A and transporting AQP2 to the apical membrane of principal cells [39,40]. AVP-mediated water retention and dilutional hyponatremia are prevalent in patients with heart failure and after cardiac surgery [41,42]. Notably, the UAQP2 level is closely associated with the AQP2 in the kidney [23,24]. Our findings are consistent with current theories regarding the pathophysiological role of AVP in cardiorenal syndrome and support the diagnostic value of UAQP2 for detecting AKI in patients with ADHF.

AKI is common in individuals with ADHF, and a traditional diagnosis of AKI through the creatinine method is limited by its 24–72 h delay from onset to elevation [13]. Previous studies have also supported the use of biomarkers to improve the diagnosis and prognosis of AKI in the critically ill population [43,44]. Clinicians can deploy nephroprotective measures to improve patient outcomes if AKI is recognized early [6]. Although some experts have expressed doubt regarding the use of biomarkers for detecting AKI in ADHF patients due to their inconsistency, the current finding of UAQP2 elevation in AKI among patients with heart failure is promising because of its potential therapeutic role. Currently, V2 receptor antagonists such as tolvaptan are commercially available and are used for patients with conditions ranging from heart failure refractory to conventional diuretics [45]. A recent study by Imamura et al. proposed using UAQP2 for prediction of the responsiveness to tolvaptan in patients with decompensated heart failure [46]. In a relatively preserved collecting duct, plasma AVP stimulates AQP2 phosphorylation and transport to the apical membrane of principal cells, and UAQP2 can be used as a functional biomarker for the collecting duct [46]. By contrast, UAQP2 is almost undetectable in patients unresponsive to tolvaptan therapy, such as patients with advanced CKD or diabetic nephropathy, possibly due to collecting duct function deterioration [46,47,48,49]. Moreover, a recent study also proposed changes in UAQP2 could be used to predict the tolvaptan response in patients with autosomal dominant polycystic kidney disease [50]. Our finding of increased UAQP2 levels in AKI patients with ADHF may help with identifying a feasible candidate for patients responsive to tolvaptan therapy. In addition, the safety profile of tolvaptan was favorable, and patients experienced only minor side effects, such as thirst and a dry mouth [51]. Further investigation is required to assess the use of UAQP2-guided tolvaptan therapy in ADHF patients with AKI.

The UAQP2 elevation in ADHF-induced AKI may not be universal in all AKI scenarios. Decreases in exosomal UAQP2 levels have been measured in several animal models of AKI, including I/R, lipopolysaccharide (LPS)-induced, cisplatin-induced, and gentamicin-induced kidney injury [30,31,32,39,52]. Asvapromtada et al. reported significantly decreased levels of exosome UAQP2 during severe AKI induced by I/R. Apical trafficking of AQP2 in the principal cell also diminishes in I/R-induced kidney injury. In addition, in unilateral I/R AKI rats, decreased urine osmolality and increased urine volume were reported, suggesting a severe urinary concentration defect [30]. Additionally, in LPS-induced AKI, expression of AQP2 is also decreased despite a marked increase in serum AVP levels [53]. One explanation for such a difference in UAQP2 levels may be the varying pathophysiology of AKI; AKI is a syndrome with heterogenous etiologies, and different AKI models have varying mechanisms of renal injury. For example, AVP plays an essential role in the pathophysiology of ADHF-related AKI but much less so in other types of AKI. In unilateral I/R AKI, an endothelin-1 increase may inhibit AVP-induced water permeability via endothelin type B receptors [30,54]. Accelerated degradation of AQP2 protein was proposed for LPS-induced AKI [53]. Another possible explanation may be different timing of the measurement of UAQP2 relative to the onset event. We measured UAQP2 when patients were admitted to the CCU, whereas decreased UAQP2 levels may be observed on day 7 in a bilateral I/R rat model. Sequential measurement of UAQP2 in patients with ADHF may provide more insight. Unlike previous studies, which measured exosomal UAQP2, we analyzed total UAQP2 in this study. AQP2 exists in both a soluble form and membrane-bound form in the urine, and UAQP2 is predominantly localized in the low-density exosome [55]. UAQP2 in the exosome is thus proportional to UAQP2 levels as a whole [20,55]. Similar to soluble UAQP2, the exosomal UAQP2 level is correlated with the renal AQP2 level [56]. ELISA was used in our study to quantify UAQP2, which measures both exosomal and soluble forms of UAQP2 with good performance [57]. Although exosomal UAQP2 measurement may be a good candidate, the optimal method for isolating exosomes is still debated [58]. Further study is required.

Our study has several limitations. First, we measured UAQP2 only once to predict AKI in ADHF patients. Sequential measurement of biomarkers may better reflect kidney injury and improve predictive power. In addition, plasma AVP levels were unavailable in our study. Second, both the roles and the expression of UAQP2 in AKI with ADHF require additional investigation. The mechanism of urinary secretion of AQP2 is still poorly understood. Further animal modeling may help to extrapolate the exact mechanism and applications of UAQP2 in ADHF with AKI. Third, urinary exosomes were not evaluated in our study due to technical limitations. Finally, additional prospective trials are warranted to explore the use of UAQP2 for AKI, given the small sample size and observational design of our study.

## 5. Conclusions

In summary, UAQP2 demonstrates acceptable discriminative power for early detection of AKI in patients with ADHF, as well as serum BNP. In addition, the combination of the two markers may serve as a novel, noninvasive biomarker to differentiate AKI. The combination had the highest AUROC; thus, it has potential in early identification of AKI. Further study is required to evaluate the relation between UAQP2 levels and tolvaptan responsiveness in AKI for patients with ADHF.

## Figures and Tables

**Figure 1 biomedicines-10-00613-f001:**
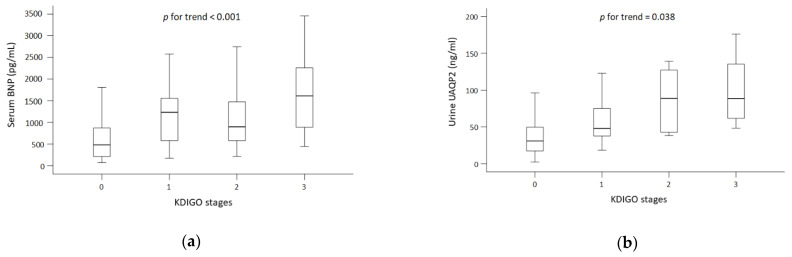
Levels of serum BNP (**a**) and urine AQP2 (**b**) across KDIGO stages. Abbreviations: AQP2, aquaporin 2; BNP, brain natriuretic peptide; KDIGO, Kidney Disease: Improving Global Outcomes. Abbreviations: BNP, brain natriuretic peptide; UAQP2, urinary excretion of aquaporin 2; UCr, urine creatinine.

**Figure 2 biomedicines-10-00613-f002:**
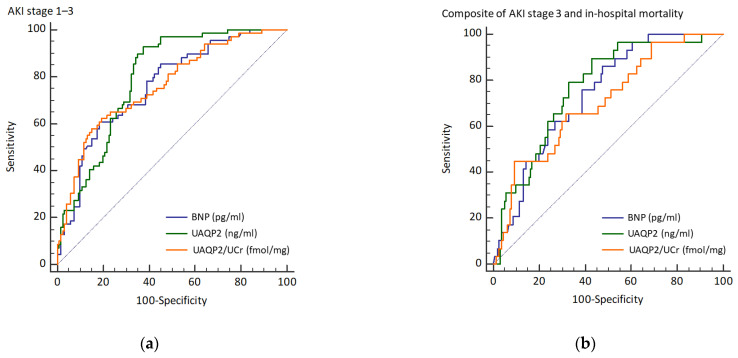
Receiver operating characteristic curves of BNP, UAQP2, and UAQP2/UCr discriminating (**a**) AKI and (**b**) composite of AKI stage 3 and in-hospital mortality. Abbreviations: BNP, brain natriuretic peptide; UAQP2, urinary excretion of aquaporin 2; UCr, urine creatinine.

**Figure 3 biomedicines-10-00613-f003:**
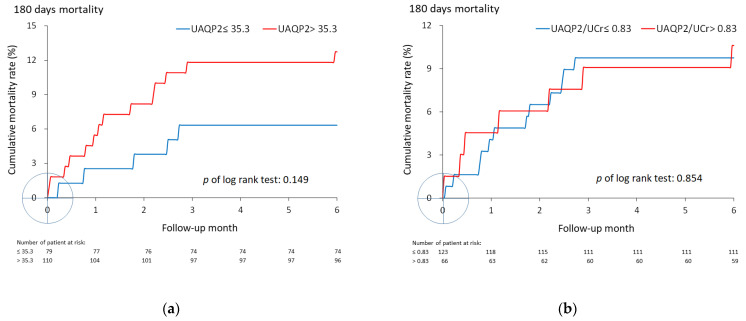
Cumulative mortality rates during 180-day follow-up of patients, with stratification of the optimal cutoff of UAQP2 (**a**) and UAQP2/UCr (**b**). Abbreviations: UAQP2, urinary excretion of aquaporin 2; UCr, urine creatinine.

**Table 1 biomedicines-10-00613-t001:** Baseline characteristics of patients admitted due to heart failure with or without AKI.

Variable	AKI+ (*n* = 69)	AKI- (*n* = 120)	*p* Value
Male, n (%)	43 (62.3)	86 (71.7)	0.197
Age (year)	70.9 ± 11.7	66.3 ± 14.4	0.024
MAP (mmHg)	89.5 ± 21.7	85.2 ± 17.2	0.137
LVEF (%)	45.1 ± 19.6	50.7 ± 17.7	0.048
Comorbidity, n (%)			
Chronic kidney disease	38 (55.1)	17 (14.2)	<0.001
Hypertension	55 (79.7)	74 (61.7)	0.014
Diabetes mellitus	40 (58.0)	55 (45.8)	0.131
Laboratory data			
Baseline creatinine (mg/dL)	1.8 ± 1.3	1.1 ± 0.6	<0.001
WBC (×103)	10.6 ± 5.6	9.6 ± 3.3	0.132
Hb (mg/dL)	11.0 ± 2.3	12.7 ± 2.4	<0.001
Glucose (mg/dL)	184 ± 77	163 ± 81	0.093
Sodium (mg/dL)	138.9 ± 4.2	138.3 ± 3.8	0.351
Potassium (mg/dL)	4.09 ± 0.67	3.85 ± 0.49	0.007
Biomarkers			
BNP (pg/mL)	1210 [639, 1740]	479 [212, 869]	<0.001
UAQP2 (ng/mL)	61.5 [41.8, 110.3]	30.9 [17.4, 49.6]	<0.001
UAQP2/UCr (fmol/mg)	1.09 [0.41, 2.18]	0.35 [0.16, 0.68]	<0.001

AKI, acute kidney injury; BNP, brain natriuretic peptide; Hb, hemoglobin; LVEF, left ventricular ejection fraction; MAP, mean arterial pressure; UAQP2, urinary excretion of aquaporin 2; UCr, urine creatinine; WBC, white blood cell count. Continuous data are presented as mean ± standard deviation or median (25th, 75th percentile).

**Table 2 biomedicines-10-00613-t002:** In-hospital outcome and readmission rate of patients admitted due to heart failure with or without AKI.

Variable	AKI+ (*n* = 69)	AKI- (*n* = 120)	*p* Value
In-hospital mortality, n (%)	8 (11.6)	5 (4.2)	0.072
AKI stage, n (%)			
Stage 1	35 (50.7)	-	
Stage 2	16 (23.2)	-	
Stage 3	18 (26.1)	-	
Composite of AKI stage 3 and in-hospital mortality	24 (34.8)	5 (4.2)	<0.001
Renal replacement, n (%)	7 (11.1)	0 (0.0)	0.001
ICU stay (days)	5.0 [3.0, 8.0]	3.0 [2.0, 5.0]	0.002
Readmission in 180 days, n (%)	15 (24.6)	15 (13.2)	0.062
Readmission in 365 days, n (%)	21 (34.4)	25 (22.1)	0.104

AKI, acute kidney injury; ICU, intensive care unit. Continuous data are presented as median (25th, 75th percentile).

**Table 3 biomedicines-10-00613-t003:** Association of BNP, UAQP2, and UAQP2/Cr with the risk of AKI and the composite of AKI stage 3 and in-hospital mortality.

	OR (95% CI)	
Variable	Any AKI Stage	Composite Outcome #
BNP, per 1000 pg/mL increase		
Model 1, unadjusted model	2.55 (1.70–3.81) *	1.73 (1.23–2.44) *
Model 2, adjusted for age, sex	2.43 (1.61–3.67) *	1.78 (1.24–2.56) *
Model 3, further adjusted for DM, HTN	2.49 (1.63–3.80) *	1.80 (1.25–2.59) *
Model 4, further adjusted for MAP, LVEF	2.36 (1.52–3.66) *	1.92 (1.29–2.86) *
Model 5, further adjusted for Hb, baseline SCr	1.87 (1.20–2.92) *	1.48 (0.91–2.41)
UAQP2, per 50 ng/mL		
Model 1, unadjusted model	2.06 (1.36–3.12) *	1.00 (0.94–1.08)
Model 2, adjusted for age, sex	2.01 (1.31–3.07) *	1.01 (0.94–1.08)
Model 3, further adjusted for DM, HTN	1.92 (1.25–2.95) *	1.01 (0.94–1.08)
Model 4, further adjusted for MAP, LVEF	1.96 (1.26–3.04) *	1.01 (0.94–1.08)
Model 5, further adjusted for Hb, baseline SCr	1.58 (1.03–2.41) *	1.00 (0.93–1.09)
UAQP2/UCr, fmol/mg		
Model 1, unadjusted model	2.48 (1.66–3.71) *	1.08 (0.96–1.23)
Model 2, adjusted for age, sex	2.48 (1.62–3.80) *	1.08 (0.95–1.23)
Model 3, further adjusted for DM, HTN	2.42 (1.58–3.71) *	1.09 (0.95–1.24)
Model 4, further adjusted for MAP, LVEF	2.41 (1.58–3.68) *	1.09 (0.95–1.24)
Model 5, further adjusted for Hb, baseline SCr	1.94 (1.29–2.92) *	1.05 (0.90–1.24)

AKI, acute kidney injury; BNP, brain natriuretic peptide; CI, confidence interval; DM, diabetes mellitus; Hb, hemoglobin; HTN, hypertension; LVEF, left ventricular ejection fraction; MAP, mean arterial pressure; OR, odds ratio; SCr, serum creatinine; UAQP2, urinary excretion of aquaporin 2; UCr, urine creatinine. * *p* < 0.05; # AKI stage 3 and in-hospital mortality.

**Table 4 biomedicines-10-00613-t004:** Discriminating between AKI and the composite of AKI stage 3 and in-hospital mortality: receiver operating characteristic curve analysis of the biomarkers of heart failure and renal dysfunction.

Outcomes/Marker	AUC, % (95% CI) †	Cut-Off #	Sensitivity, % (95% CI)	Specificity, % (95% CI)
AKI stage 1–3				
BNP (pg/mL)	75.9 (69.0–82.9) *	>950.4	60.9 (48.4–72.4)	81.7 (73.6–88.1)
UAQP2 (ng/mL)	79.5 (73.2–85.7) *	>35.3	92.8 (83.9–97.6)	62.5 (53.2–71.2)
UAQP2/UCr (fmol/mg)	76.1 (69.1–83.2) *	>0.83	62.3 (49.8–73.7)	80.8 (72.6–87.4)
BNP+ UAQP2	80.2 (74.1–86.4) *	NA	NA	NA
BNP + UAQP2/UCr	80.2 (73.9–86.5) *	NA	NA	NA
Composite of AKI stage 3 and in-hospital mortality				
BNP (pg/mL)	73.9 (65.5–82.3) *	>618	86.2 (68.3–96.1)	52.5 (44.5–60.4)
UAQP2 (ng/mL)	76.6 (68.1–85.0) *	>38.5	89.7 (72.6–97.8)	56.9 (48.8–64.7)
UAQP2/UCr (fmol/mg)	70.2 (60.1–80.3) *	>1.88	44.8 (26.4–64.3)	90.6 (85.0–94.7)
BNP+ UAQP2	73.1 (64.1–82.1) *	NA	NA	NA
BNP + UAQP2/UCr	74.9 (66.8–83.0) *	NA	NA	NA

AKI: acute kidney injury; AUC, area under curve; BNP, brain natriuretic peptide; CI, confidence interval; NA, not applicable; UAQP2, urinary aquaporin 2; UCr, urine creatinine. † analyzed using DeLong’s test. # according to the Youden index. * *p* < 0.05.

## Data Availability

The data presented in this study are available on request from the corresponding author. The data are not publicly available due to privacy/ethical restriction.

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
