# Peer review of "Predictive Value of Urinary Aquaporin 2 for Acute Kidney Injury in Patients with Acute Decompensated Heart Failure"

_biomedicines, 2022, doi:10.3390/biomedicines10030613_

Round 1

Reviewer 1 Report

The paper of Chan and colleagues: “Predictive Value of Urinary Aquaporin 2 for Acute Kidney In- 2

jury in Patients with Acute Decompensated Heart Failure” presents the results of the innovative and interesting study in patients with ADHF.

The manuscript perfectly fits in with quite a number of publications related to the role of urinary/serum biomarkers in chronic disorders.
The Abstract is well written, accurate, intelligible on its own.
The Introduction includes the aim of the study and review of the key literature.
The Results and methods sections are clear. Tables and Figures are accurate and accessible. Results and Methods provide all necessary details to support Authors conclusions.
Discussion is well written and presents the main conclusions, compare the obtained results with the avaliable literature. Moreover, the Authors describe the limitations of their study and plans for future research.
To sum up, the value of the article is great. The paper highlights an interesting topic, and is innovative.
The manuscript is written in a good style and easy to read, there are no grammatical, punctuation or linguistic errors. In the manuscript there are good, clear figures and tables that facilitate understanding of the issues described in the text.
The paper is valuable to publish in Biomedicines.

Author Response

Thank you for your comment

Reviewer 2 Report

Authors analyzed UAQP-2 and sBNP in patients with ADHF and confirm their usefulness (each or in combination) as early biomarkers of AKI in these cases. They  proposed UAQP-2 as possible predictor to therapy with V2 antagonist tolvaptan.  

row 23 abbreviation "AUROCs" have to be explain. 

Early diagnosis of AKI in patients with ADKD is challenging, because creatinine method is imperfect due to 1-3 days delay in its elevation form insult. Several biomarkers can be raised in serum or urine due to glomerular or tubular injury.  AQP-2 is interesting because appears in urine (UAQP-2) in patients with ADCD due to vasopressin stimulation of V2 receptor in the apical site of collecting duct and AQP-2 expression. In this sample  were included 189 patients admitted to he Coronary Unit from 2009-2014. Biomarker UAQP-2 was measured with ELISA method only once, within first 7 days of admission and correlated with demographic and clinical data, sBNP and sCr, as well  as survival. Combination of sBNP and UAQP-2, as well as UAQP-2/UCr was correlated with AKI grade, and has an impact on survival in AKI grade 3. Authors conclude that UAQP-2 is good early biomarker of AKI in patients with ADKD and potential predictor for therapy with V2 antagonist tolvaptan

Author Response

Thank you for your comment

We also revised the manuscript to clarify AUROC on row 23 accordingly.

Reviewer 3 Report

The manuscript is well written and understandable, I have only minor concerns for the authors. The enrolment period was between 2009 and 2014 with 1 year of follow-up. The Authors should explain why they have submitted the manuscript In 2022.

The bibliography should be enlarged and modified with some more news articles. The manuscript presents no bibliography from 2022 or 2021, only 3 from 2020 and 2019 with over 50 articles cited in the text.

Author Response

Dear Reviewer:

Thank you for your comment.

We have updated the citation to include most updated literature. However, in the most updated search, only 55 articles were published during 2021 to 2022 regarding urinary aquaporin 2. All relevant studies were cited in this revision. Other study regarding urinary aquaporin 2 only include small patient number and not focusing on acute kidney injury. (such as reference 42 and 50). Although urinary aquaporin 2 have been discovered for a long time, its implication in acute kidney injury is not well described and it is the strength of our study.

Our study was conducted between 2009 and 2014 with 1 year of follow-up. However, we spent some time to test the reliable ELISA kit to measure Urine aquaporin 2. We also tried to develop self-synthesized antibody for urinary aquaporin 2. In addition, clinical work due to recent pandemic postpones our submission. Although our late submission, there is no literature describing urinary aquaporin 2 in acute kidney injury patients with acute decompensated heart failure. We are sincerely sorry about the delay.